# How do Query Features correlate with Runtime? An evaluation of SPARQL Querying Interfaces

Hashim Khan[1], Muhammad Saleem[1] and Axel-Cyrille Ngonga Ngomo[1]

[1]*Data Science Group (DICE), Heinz Nixdorf Institute, Paderborn University, Germany*

## Abstract

To address the recent challenge of online availability and reliability of public SPARQL Endpoints, various SPARQL querying interfaces based on client-server architectures have been developed. These interfaces aim to achieve a balanced distribution of query execution workload between the server and the client, thus improving service availability, scalability, and runtime performance. A central aspect of this workload distribution is query planning, which is significantly influenced by various SPARQL query features, including the number of triple patterns, join vertices, projection variables, and the result set size. Understanding how these query features affect performance is essential for the effective design of such interfaces. In this study, we conduct a performance evaluation of several well-known SPARQL querying interfaces to investigate the relationship between query structure and runtime performance. Through extensive experiments, we analyze how different query features correlate with execution time, and which interfaces are best suited for specific query types. Our findings provide valuable insights into the behavior of these interfaces under different structural conditions. To the best of our knowledge, this work represents the first systematic attempt to assess the correlation between SPARQL query features and runtime performance across a diverse set of client-server-based querying interfaces. We expect our results to support the community in designing more effective query planning algorithms and optimizing interface behavior to enhance both runtime efficiency and service availability in decentralized RDF data environments.

## Keywords

SPARQL features, Querying interfaces, Runtime, Knowledge graph

## 1. Introduction

The continuous growth of Knowledge Graphs (KGs) on the Web raises new challenges for querying and integrating very large amounts of data across multiple sources. Efficient access to these data remains essential for the scalability of Linked Data and Semantic Web technologies. To fully exploit the potential of such data, users must be able to query and combine it in a simple, effective, and efficient way. In practice, RDF knowledge graphs are typically accessed through two broad classes of architectures: (1) SPARQL endpoints, in which data is stored on the server in a triple store and the full query is executed on the server side; and (2) SPARQL querying interfaces, in which query processing is distributed between client and server.

Multiple studies [1, 2, 3, 4, 5, 6] have examined the performance of various RDF triple stores. A more exhaustive list of the triple stores and evaluation studies is reported in [7]. While SPARQL endpoints have been widely studied in the literature [7], this work focuses on SPARQL querying interfaces and on how their performance relates to SPARQL query features.

SPARQL querying interfaces were introduced to address several limitations of traditional endpoints, including limited scalability, high server load, and poor responsiveness under concurrent access. Interfaces such as Triple Pattern Fragments (TPF) [8], Bindings restricted TPF (brTPF) [9], Web Preemption for Public SPARQL Query Services (SaGe) [10], Star Pattern Fragments (SPF) [11], Hybrid Shipping for SPARQL Querying on the Web (Smart-KG) [12], and Balanced Access to Web Knowledge Graphs (Wise-KG) [13] have been proposed to address these issues. These interfaces follow different execution

*UKG 2026: 2nd International Workshop on Users and Knowledge Graphs, September 15, 2026, Ghent, Belgium (co-located with SEMANTiCS 2026)*

*Corresponding author: Hashim Khan.

✉ hashim.khan@uni-paderborn.de (H. Khan); saleem@mail.upb.de (M. Saleem); axel.ngonga@upb.de (A. N. Ngomo)

🆔 0009-0006-5953-480X (H. Khan); 0000-0001-9648-5417 (M. Saleem); 0000-0001-7112-3516 (A. N. Ngomo)

models, including triple pattern decomposition, client side joins, server preemption, and hybrid execution plans, each with its own performance trade-offs.

Several studies [12, 13, 11, 10, 9] have compared these interfaces, either with SPARQL endpoints or among themselves. However, no unified evaluation exists that examines all prominent interfaces under identical benchmark settings. Moreover, most prior work focuses on absolute performance measures such as execution time or query completeness, without explaining the reasons for performance variation across queries.

One important factor is the structure of the query itself. Saleem et al. [14] showed that SPARQL query features such as the number of triple patterns, join vertices, projection variables, and result set size can significantly affect runtime. Their study, however, was limited to SPARQL endpoints. To the best of our knowledge, no existing study has quantified the relationship between query features and runtime across the full range of SPARQL querying interfaces.

This paper addresses these critical gaps. First, we conduct a comprehensive runtime evaluation of six widely used SPARQL querying interfaces, namely TPF, brTPF, SaGe, SPF, Smart-KG, and Wise-KG, using the WatDiv SPARQL benchmark on two dataset scales, with 10 million and 100 million triples. Second, we analyze the statistical correlation between key SPARQL query features and runtime across all interfaces using Spearman's rank correlation coefficient. Together, these analyses help explain not only how well each interface performs, but also why it performs that way. Our findings provide insight into the design and tuning of SPARQL querying systems, as well as practical guidance for users who write SPARQL queries.

Our main contributions are as follows.

- We present a comprehensive runtime evaluation of six widely used SPARQL querying interfaces. The experiments are based on the WatDiv [3] benchmark with two dataset sizes, 10 million and 100 million triples, offering insight into how performance scales with dataset size.
- We systematically investigate the correlation between SPARQL query features and runtime across all evaluated interfaces using Spearman's rank correlation coefficient. This gives a statistically grounded view of which features have the strongest influence on runtime in different querying models.
- We provide a unified performance comparison that reveals both the strengths and the bottlenecks of each interface under different query structures and dataset scales.
- Our evaluation shows how different execution models respond to structural query complexity, offering useful evidence for future optimization strategies in SPARQL engines and Web based query services.

The rest of the paper is organized as follows. Section 2 introduces the SPARQL querying interfaces selected for evaluation. Section 3 presents the SPARQL query features used in our analysis. Section 4 summarizes related work and identifies the research gap addressed in this paper. Section 5 describes the experimental setup, benchmarks, and queries. Section 6 presents the results and discussion. Section 7 describes the online availability of the resources used in this study and how to reproduce our results. Finally, Section 8 concludes the paper and outlines future directions.

## 2. SPARQL Querying Interfaces

This section provides a detailed description of the six SPARQL querying interfaces evaluated in our study. Each interface represents a distinct approach to addressing the scalability and availability challenges of traditional SPARQL endpoints by redistributing query processing workload between client and server.

- **Triple Pattern Fragments (TPF):** TPF [8] constitutes a foundational lightweight interface that fundamentally shifts the majority of query processing responsibility from server to client. The server handles only simple triple pattern requests, returning paginated result sets along with metadata such as total count estimates and hypermedia controls for navigation. All complex

operations—joins, filtering, ordering, aggregation, and result projection—are delegated to the client. This design ensures minimal server complexity, bounded response times, and high availability even under concurrent load, as each request requires only index lookups without intermediate result materialization. However, TPF incurs substantial network overhead, as clients must retrieve large volumes of intermediate triples to perform joins locally, leading to numerous HTTP requests and high data transfer costs, particularly for multi-join queries with exponentially growing intermediate result sizes. For our experiments, we used the JavaScript-based TPF server[1] together with the Comunica client framework [15][2].

- **Bindings-restricted Triple Pattern Fragments (brTPF):** brTPF [9] extends TPF by enabling clients to include intermediate join bindings with triple pattern requests. The server then returns only those triples from the requested pattern that contribute to joins with the provided bindings, effectively implementing selective filtering at the server side. This mechanism significantly reduces network traffic, while maintaining the same low server complexity and availability guarantees. The client retains responsibility for join coordination and final result assembly but receives substantially less data per request. Despite these improvements, brTPF still faces challenges with queries generating large intermediate result sets and requires sophisticated client-side query planning to maximize its benefits.

- **Web Preemption for Public SPARQL Query Services (SaGe):** SaGe [10] introduces web preemption as a mechanism for managing long-running queries in public SPARQL services. The server allocates fixed time slices (typically 75ms) to each query, suspending execution when the quantum expires and returning partial results together with serialized query execution state. Clients can resume processing by sending continuation requests with the preserved state. This approach prevents individual queries from dominating server resources, improves overall system responsiveness under concurrent load, and achieves higher query completion rates than TPF/brTPF in high-concurrency scenarios. However, SaGe introduces server-side overhead from frequent context switching and state serialization, supports only basic SPARQL operators (leaving complex features to clients), and requires clients to manage resumption logic and execution state reliably.

- **Star Pattern Fragments (SPF):** SPF [11] extends the TPF model by supporting star-shaped query patterns, where multiple triple patterns share a common subject, within a single request. The server evaluates the complete star pattern by performing joins between predicates of that subject and returns fully joined results for the entity. Compared to TPF, this approach significantly reduces both HTTP requests and intermediate data volume by exploiting the locality inherent in RDF data models, where entities typically exhibit star-shaped access patterns. While clients still coordinate joins between different stars (corresponding to different subjects), SPF eliminates client-side joins within individual entities. The approach proves particularly effective for entity-centric exploration queries that retrieve multiple properties of focal resources.

- **Hybrid Shipping for SPARQL Querying on the Web (Smart-KG):** Smart-KG [12] introduces partition-based query processing as a hybrid extension to TPF. The server organizes RDF data into characteristic sets, which are groups of entities sharing identical predicate sets, and compresses these into Header-Dictionary-Triples (HDT)[3] partitions. Clients access metadata about partition selectivity and size to choose between conventional TPF-style triple pattern requests or fetching entire partitions for local processing. For star-shaped sub-queries that match partition predicates exactly, clients receive compact, self-contained data units that eliminate multiple round trips and enable efficient local evaluation. When partitions do not align with query patterns, Smart-KG falls

---

[1]TPF Server: https://github.com/LinkedDataFragments/Server.js
[2]Comunica client: https://github.com/comunica/comunica
[3]HDT: https://www.rdfhdt.org/what-is-hdt/

back to individual triple pattern retrieval. This adaptive strategy substantially reduces network traffic and server load for suitable queries, although it requires significant client-side storage and decompression capabilities.

- **Balanced Access to Web Knowledge Graphs (Wise-KG):** Wise-KG [13] represents the current state of the art by combining Smart-KG's partition shipping with SPF's server-side star pattern processing. A dynamic cost model considers multiple factors including server CPU load, network bandwidth, client capabilities, partition statistics, and estimated data transfer costs to determine the optimal execution strategy for each star-shaped sub-query. Rather than relying on static heuristics, Wise-KG decomposes queries into star patterns, estimates their cardinalities, and assigns each sub-query to either partition shipping for client-side HDT evaluation or SPF processing for server-side star evaluation, with the goal of minimizing total execution time. Experimental results show up to 4x performance improvements over SPF and Smart-KG on large datasets, with better load balancing and fewer timeouts. However, the approach remains optimized for star-shaped patterns, introduces runtime overhead from cost estimation, and requires accurate modeling of heterogeneous system resources.

## 3. SPARQL Query Features

The literature on SPARQL query benchmarking [16, 17, 18, 19, 20] consistently emphasizes the need for comprehensive benchmarks that capture the full structural and functional diversity of real-world SPARQL queries. Essential characteristics include variability in the number of triple patterns, projection variables, join vertices, result set sizes, query execution times, and basic graph pattern (BGP) complexity. A robust benchmark must also account for selectivity constraints on triple patterns and join vertices, diverse join topologies (linear, star, snowflake), and the presence of frequently used SPARQL clauses such as `LIMIT`, `OPTIONAL`, `ORDER BY`, `DISTINCT`, `UNION`, `FILTER`, and `REGEX`. These factors collectively determine query performance in practical deployment scenarios [14].

For our analysis, we selected four key SPARQL query features that previous studies [14] identified as having the strongest influence on execution time across SPARQL endpoints: (1) *Number of Projection Variables (PV)*, (2) *Number of Join Vertices (JV)*, (3) *Number of Triple Patterns (TP)*, and (4) *Result Set Size (RS)*. These features were chosen based on their demonstrated statistical significance and practical relevance to query optimization.

```
SELECT * WHERE {
    ?drug :description ?drugDesc .     // TP1
    ?drug :drugType :smallMolecule .   // TP2
    ?drug :keggCompoundId ?compound .  // TP3
    ?enzyme :xSubstrate ?compound .    // TP4
    ?chemReac :xEnzyme ?enzyme .       // TP5
    ?chemReac :equation ?chemEq .      // TP6
    ?chemReac :title ?reacTitle        // TP7
}
```

**Figure 1:** An example SPARQL query

Figure 1 illustrates these features using a representative biomedical query. This query contains seven *projection variables* (∗ selects all bound variables), four *join vertices* (shared variables between triple patterns), and seven *triple patterns*. The *result set size* (RS) depends on actual execution against a specific dataset.

Each feature influences runtime through distinct mechanisms. A higher number of projection variables (PV) increases network transfer costs, as more data must be serialized and transferred to complete the result set. More triple patterns (TP) typically increase query complexity, requiring additional join operations whose cost depends critically on join ordering and intermediate result sizes. The number

of join vertices (JV) reflects the query's connectivity, with star and snowflake patterns often exhibiting different scaling behavior than linear chains. Finally, larger result set sizes (RS) increase both computation and transfer overhead, particularly in distributed settings where intermediate results must be materialized and joined across multiple sources.

These dependencies make query runtime prediction challenging but critical for optimization. Understanding feature-runtime relationships across diverse querying interfaces enables more effective query planning, source selection, and adaptive execution strategies, which form the foundation for our subsequent evaluation.

## 4. Related Work

Our work builds on research evaluating knowledge and data management systems within both the database and Semantic Web communities. Numerous benchmarks [3, 4, 5, 2, 6] have been developed to assess RDF triple stores and SPARQL query engines, as comprehensively surveyed by Ali et al. [7]. While SPARQL endpoints have received substantial attention, evaluations of SPARQL querying interfaces remain more limited.

Several studies have compared individual interfaces using WatDiv [3] as benchmark. Hartig et al. [9] compared brTPF and TPF through network usage metrics (request counts, triple volumes). Minier et al. [10] evaluated SaGe against Virtuoso and brTPF measuring throughput, timeouts, and result completeness. Christian et al. [11] assessed SPF against TPF and brTPF using request counts, execution time, transferred bytes, and CPU load. Azzam et al. [12, 13] measured Smart-KG and Wise-KG against TPF, Virtuoso, and SaGe focusing on execution time and resource consumption. These works provide valuable insights but lack unified evaluation comparing all prominent client-server interfaces under identical conditions.

Recent research has also investigated AI applications in SPARQL processing. Zhang et al. [21] developed SVR and $k$-NN models to predict execution time, CPU, and memory usage from algebraic query features. Casals et al. [22] proposed tree-based convolutional neural networks that exploit query structure to achieve higher prediction accuracy. Eschauzier et al. [23] presented a reinforcement-learning approach to join order optimization that uses Tree-LSTM networks for latency estimation. Advances in large language models have substantially improved the translation of natural language questions into SPARQL queries [24]; Diallo et al. [25] further strengthened generation quality by integrating copy mechanisms and semantic-similarity based example selection. Methods such as RDF2Vec [26] and AutoRDF2GML [27] enable the application of graph neural networks to downstream semantic-web tasks, although their primary focus remains knowledge discovery rather than query execution optimization.

However, existing AI approaches exhibit fundamental limitations for client-server based querying architectures. They focus on narrow tasks like performance prediction or join ordering without addressing workload distribution, the core challenge of client-server interfaces. Feature representations remain limited to traditional query characteristics, neglecting client capacity, network latency, and server load dynamics. Evaluations target server-centric endpoints, ignoring distributed constraints like network minimization and client-server coordination.

Our work fills this gap through comprehensive runtime evaluation of six SPARQL querying interfaces and statistical analysis of query feature correlations, providing empirical foundations for future AI-driven workload optimization across hybrid architectures.

## 5. Evaluation Setup

Evaluating RDF based interfaces requires including an RDF dataset, SPARQL queries, and performance metrics. These components are vital for conducting a thorough and accurate assessment of research contributions, as highlighted in prior studies [28].

***RDF Dataset.*** In this study, we used the *Waterloo SPARQL Diversity Test Suite (WatDiv)*, a comprehensive benchmark comprising a synthetic data generator and a query generator. The data generator offers the flexibility to produce RDF data with adjustable structuredness parameters, while the query generator can generate queries based on various query templates [29]. Synthetic data benchmarks, as demonstrated in prior research [30], hold significant value for assessing system scalability across datasets of varying sizes.

For our experimentation, we selected two datasets, one comprising 10 million triples and another containing 100 million triples. These diverse datasets allow researchers to evaluate system performance across varying data loads. Detailed information on these datasets is provided in Table 1.

**Table 1**
Details about the dataset structure.

| Dataset Type | Benchmark/Dataset | # Triples | # Subjects | # Predicates | # Objects |
| --- | --- | --- | --- | --- | --- |
| Synthetic | WatDiv-10-Million | 10,916,457 | 521,585 | 86 | 1,005,832 |
| Synthetic | WatDiv-100-Million | 108,997,714 | 5,212,385 | 86 | 9,753,266 |

***SPARQL Queries.*** To evaluate query performance, we selected SPARQL queries based on key structural features identified in our earlier analysis. From the LSQ-enriched set of generated queries, we chose 30 queries each by varying the number of triple patterns and projection variables, with values ranging from 1 to 10. Likewise, we selected 30 queries each for different join vertex counts (ranging from 1 to 6) and result set sizes (in thousands, ranging from 1 to 10).

All selected queries were executed on datasets containing 10 million and 100 million triples. We report the results using average query runtimes to account for variability across executions. The complete details of the selected queries, based on the number and type of features they have, are available on the GitHub page linked in Section 7.

***Performance Metrics.*** Based on previous benchmarking and performance evaluations of RDF systems [31, 5, 32, 2, 6, 33], we can categorize the performance metrics for such comparisons into four primary categories:

- **Query Processing Related:** In this category, the most important performance metric is related to the query processing capabilities of RDF query engines. The query execution time holds significant importance. However, due to the large number of queries in the benchmark, reporting the execution time for individual queries may be impractical. As a result, Query Mix per Hour (QMpH) and Queries per Second (QpS) have emerged as essential performance measures to assess any RDF system's querying capabilities [2, 32, 5].

- **Data Storage Related:** RDF query engines need to process the given RDF data by creating indices before they can effectively execute queries. In this context, the key performance parameters within this category include the time required for data processing, the amount of storage space used, and the size of the generated indices [6, 5].

- **Result Set Related:** To compare two systems, it is essential that they produce identical results. Hence, assessing result set correctness and completeness becomes crucial in evaluating these systems [5, 2, 6].

- **Parallelism with/without Updates:** In some of the previous performance evaluations [34, 6, 5], they also test how well the engines can manage multiple queries running at the same time. This is done by simulating workloads involving several querying agents, with and without making changes to the dataset.

Our research focuses on assessing the runtime performance of all the mentioned querying interfaces. To achieve this, we have specifically chosen the first category of evaluation metrics, namely, *query*

*runtime in seconds.* Additionally, our experiments involve a small number of selected queries, allowing us to readily record the runtime for each individual query.

***Benchmark Execution.*** To evaluate the runtime performance of each SPARQL querying interface, we carried out the experiments in a controlled client server setting that reflects a realistic query execution environment. In this setup, every interface was deployed on a local server machine, while all queries were issued from a separate client machine. This arrangement allowed us to ensure that the evaluation conditions remained stable across all experiments and reduced the influence of external factors such as unpredictable network delays, background traffic, or variations in system load.

For each querying interface, we loaded two RDF datasets on the server side, containing 10 million and 100 million triples, respectively. Before starting the benchmark runs, we verified the successful loading of each dataset by executing SPARQL COUNT queries and confirming that the expected number of triples was present. This preliminary validation step ensured that all systems were evaluated on complete and correctly loaded data.

To avoid the influence of query concurrency, overlapping execution, or caching effects across different runs, queries were executed sequentially, one at a time, from the client. For each query, we measured the execution time in seconds using client side timing, which provided a consistent basis for comparing the runtime behavior of the interfaces. The same procedure was repeated for all query sets on both dataset sizes, ensuring that the results were directly comparable across systems and workloads.

The query sets were designed to cover a diverse range of SPARQL structural properties, including the number of join vertices, projection variables, triple patterns, and result set sizes. By organizing the queries according to these characteristics, we were able to investigate how structural complexity influences runtime behavior across different interfaces and dataset scales. This grouping also enabled us to identify whether certain query features have a stronger impact on performance than others, which is essential for understanding the practical behavior of each querying approach.

The average runtime for each query group, categorized by feature values, is presented in Figures 2 and 3 for the 10M and 100M datasets, respectively. Together, these measurements provide a systematic basis for analyzing performance trends and comparing the behavior of the interfaces under varying data volumes and query structures. The resulting evaluation supports a more detailed understanding of how interface design and query characteristics jointly affect runtime performance.

***Hardware.*** The experiments were conducted using separate client and server machines to accurately reflect real world deployment scenarios. The server machine, hosting all SPARQL querying interfaces and RDF datasets, was equipped with an Intel Core i5 8250U CPU featuring 8 physical cores with hyper threading, providing 16 logical cores. This server configuration included 8 GB of RAM and ran Ubuntu 20.04.2 LTS as the operating system.

The client machine, responsible for issuing queries and measuring execution times, featured a comparable hardware setup to ensure fair evaluation conditions across all interfaces. Both machines were connected through a controlled local network environment with minimal latency, eliminating external network variability as a confounding factor. This client server separation allowed us to capture realistic performance characteristics while maintaining reproducible experimental conditions across all benchmarking runs.

## 6. Results and Discussion

***Dataset Size vs Runtime.*** Figure 4 provides a comprehensive visualization of total runtime performance across all six SPARQL querying interfaces when processing both the 10 million and 100 million triple datasets. The y-axis displays cumulative execution time in seconds required to complete each query set through individual interfaces, with query categories organized by their primary structural characteristics and labeled as (a) projection variables (PV), (b) join vertices (JV), (c) triple patterns (TP), and (d) result set sizes (RS). This aggregate measurement captures the complete workload processing time for each

**Table 2**
Percentage difference in runtime when scaling from *WatDiv-10-Million* to *WatDiv-100-Million* dataset size.

| Interface | TP | RS | JV | PV | Avg. |
|-----------|------|------|-----|------|------|
| TPF | 308 | 129 | 224 | 222 | 220 |
| brTPF | 921 | 541 | 1245 | 948 | 913 |
| SaGe | 1557 | 818 | 848 | 1290 | 1128 |
| SPF | 1377 | 1101 | 598 | 3244 | 1580 |
| Smart-KG | 700 | 636 | 605 | 407 | 587 |
| Wise-KG | 270 | 705 | 361 | 477 | 453 |
| **Overall** | **855** | **655** | **646** | **1098** | – |

**Table 3**
Spearman's rank correlation coefficients between query features and query runtimes. Query features are abbreviated as PV: Projection Variables, JV: Join Vertices, TP: Triple Patterns, RS: Result set Size. Correlations and colors (−+): 0.00…0.19 very weak (●●), 0.20…0.39 weak (●●), 0.40…0.59 moderate (●●), 0.60…0.79 strong (●●), 0.80…1.00 very strong (●●).

| Interfaces | WatDiv-10M | | | | WatDiv-100M | | | |
|------------|------|------|------|------|------|------|------|------|
| | PV | TP | RS | JV | PV | TP | RS | JV |
| TPF | 0.84 | 0.94 | 0.55 | 0.83 | 0.71 | 0.87 | 0.76 | 0.60 |
| brTPF | 0.85 | 0.97 | 0.95 | 0.88 | 0.93 | 0.92 | 0.90 | 0.77 |
| SaGe | 0.78 | 0.67 | 0.68 | 0.77 | 0.62 | 0.88 | 0.59 | 0.88 |
| SPF | 0.66 | 0.66 | 0.97 | 0.77 | 0.76 | 0.74 | 0.72 | 0.77 |
| Smart-KG | 0.75 | 0.93 | 0.97 | 0.71 | 0.76 | 0.60 | 0.59 | 0.77 |
| Wise-KG | 0.77 | 0.66 | 0.85 | 0.83 | 0.77 | 0.76 | 0.90 | 0.71 |
| Overall | 0.78 | 0.81 | 0.80 | 0.80 | 0.76 | 0.80 | 0.74 | 0.75 |

interface under varying data volume conditions.

A fundamental observation emerges from this analysis: runtime performance scales approximately linearly with dataset size across all interfaces and query categories. This expected scalability behavior confirms that execution cost remains fundamentally tied to data volume, providing a baseline for evaluating relative interface efficiency under realistic deployment scenarios. The proportional increase observed across all query types demonstrates the robustness of our experimental design in capturing genuine performance characteristics rather than dataset specific artifacts.

For quantitative analysis of this scalability effect, Table 2 presents the percentage runtime increase between the 10M and 100M datasets across all query categories and interfaces. The projection variable (PV) query set exhibits the most dramatic scalability impact, with a 1098% runtime increase representing over tenfold performance degradation when scaling from moderate to large data volumes. This substantial difference suggests that PV intensive queries suffer disproportionately from dataset expansion, likely due to increased intermediate result materialization and binding generation costs at larger scales.

On the other hand, both join vertex (JV) and result set size (RS) query categories demonstrate greater runtime stability, exhibiting nearly identical percentage differences across dataset sizes. These query types appear less sensitive to data volume scaling, suggesting more efficient handling of intermediate computation and result transmission in the respective interfaces. The triple pattern (TP) query set occupies an intermediate position with an 855% runtime increase, representing moderate scalability sensitivity that falls between the extreme PV behavior and the more stable JV/RS categories.

The final column of Table 2 reveals particularly interesting interface specific scalability characteristics. Despite its consistently poor absolute performance documented in Figures 2 and 3, TPF demonstrates remarkable runtime stability across dataset scales, exhibiting the smallest average percentage increase among all interfaces. This noteworthy result highlights TPF's predictable scalability properties, making it

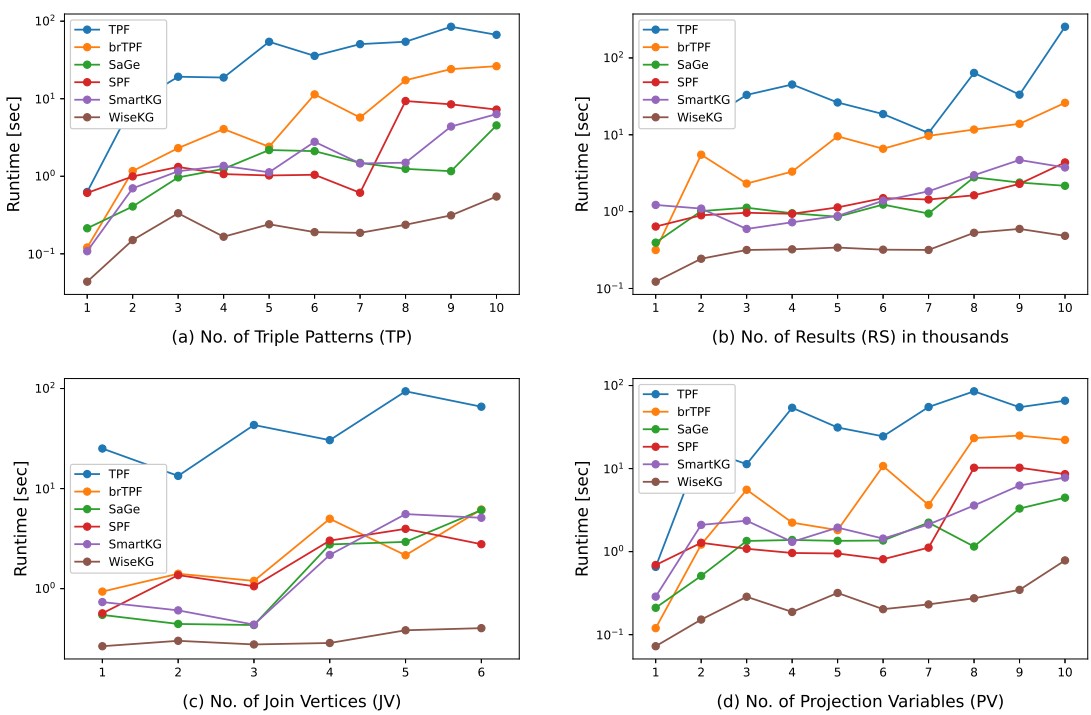

**Figure 2:** Average runtime observed by all six querying interfaces using the *WatDiv-10-Million* dataset. The x-axis represents the number of query features shown in (*a*), (*b*), (*c*), and (*d*), while the y-axis represents the runtime in seconds on a logarithmic scale.

potentially suitable for scenarios where runtime consistency outweighs absolute performance. Conversely, SPF experiences the most pronounced scalability degradation, suggesting architectural limitations that amplify under large data volumes. These divergent scalability profiles underscore the importance of considering both absolute performance and scale resilience when selecting querying interfaces for production deployment.

***Query Features vs Runtime.*** A central objective of our analysis was to systematically investigate the relationship between SPARQL query structural characteristics and their corresponding runtime performance across the evaluated querying interfaces. To quantify these relationships, we computed Spearman's rank correlation coefficient (SPC) between each query feature and the average runtime values for all six interfaces across both dataset sizes (10M and 100M triples). The complete correlation results, presented in Table 3, employ color coding to visually distinguish correlation strengths, with deeper colors indicating stronger relationships. Notably, all computed SPC values are positive, confirming that increased query complexity consistently corresponds to longer execution times across all interfaces and conditions.

The correlation analysis reveals both consistent patterns and interesting variations when comparing results across dataset scales. Across all query features and interfaces, we observe predominantly strong to very strong positive correlations, demonstrating that our selected structural characteristics—join vertices (JV), projection variables (PV), triple patterns (TP), and result set sizes (RS)—serve as reliable predictors of runtime performance. This uniformity underscores the fundamental relationship between query complexity and execution cost in SPARQL querying interfaces.

However, meaningful variations emerge when examining interface specific behaviors and dataset scale effects. For instance, Smart-KG demonstrates a very strong correlation (0.97) between result set size (RS)

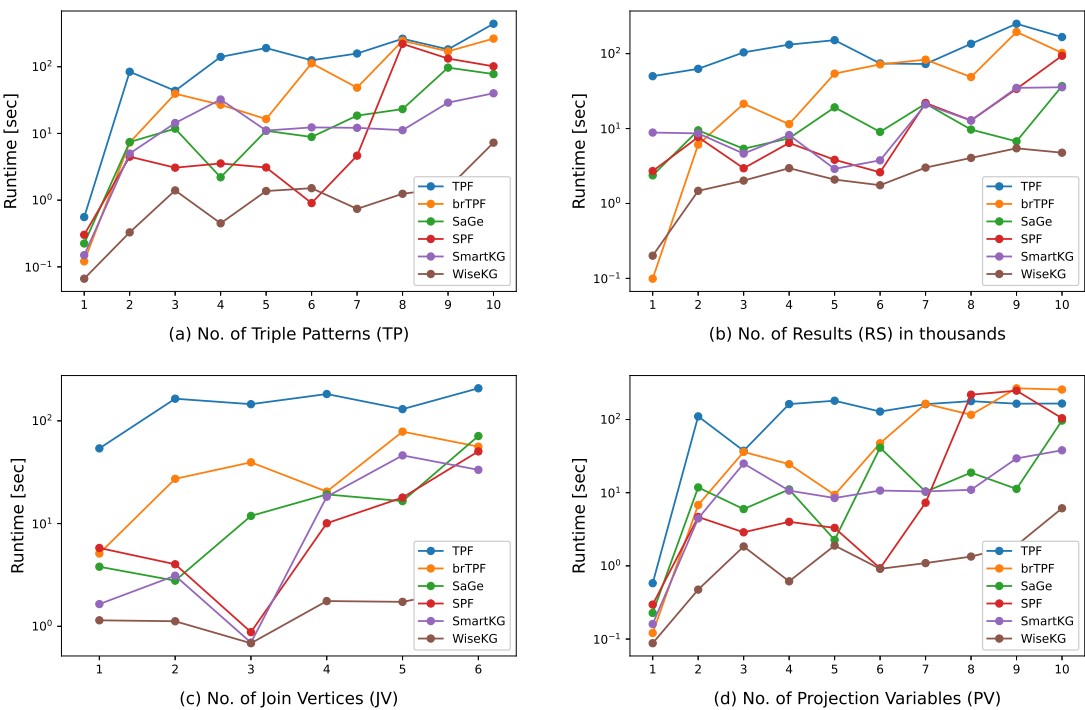

**Figure 3:** Average runtime observed by the six querying interfaces using the *WatDiv-100-Million* dataset. The x-axis represents the number of query features shown in (*a*), (*b*), (*c*), and (*d*), while the y-axis represents the runtime in seconds on a logarithmic scale.

and runtime on the WatDiv-10M dataset, indicating that output volume serves as an excellent runtime predictor for this interface under moderate data conditions. In contrast, the same relationship weakens to a moderate correlation (0.59) on the WatDiv-100M dataset, suggesting that other factors such as data volume and join complexity become more dominant at larger scales. This pattern exemplifies how the predictive power of individual query features can vary depending on both the specific querying interface architecture and the underlying dataset characteristics.

The "Overall" row in Table 3, which presents average SPC values across all interfaces for each dataset size, provides the most comprehensive perspective on feature importance. These aggregate measures confirm that all four query features maintain high correlation levels with runtime performance, with no single feature emerging as universally dominant. This balanced correlation profile validates our feature selection strategy and demonstrates that runtime behavior emerges from the complex interplay of multiple structural characteristics rather than being driven by any single dominant factor. These findings establish a solid empirical foundation for understanding query performance sensitivity and inform future optimization strategies for SPARQL querying interfaces.

***Total Runtime.*** Figure 4 presents the aggregate runtime measurements in seconds for all six SPARQL querying interfaces across both dataset sizes (10 million and 100 million triples), capturing the complete execution of all query sets. The results reveal remarkably consistent performance trends across all four query categories, demonstrating that relative interface rankings remain stable regardless of query structure or data volume.

Among the evaluated systems, TPF consistently exhibited the poorest performance across both datasets, establishing it as the slowest querying interface in our comprehensive evaluation. In stark contrast, Wise-KG achieved the best overall performance, demonstrating superior efficiency across all experimental

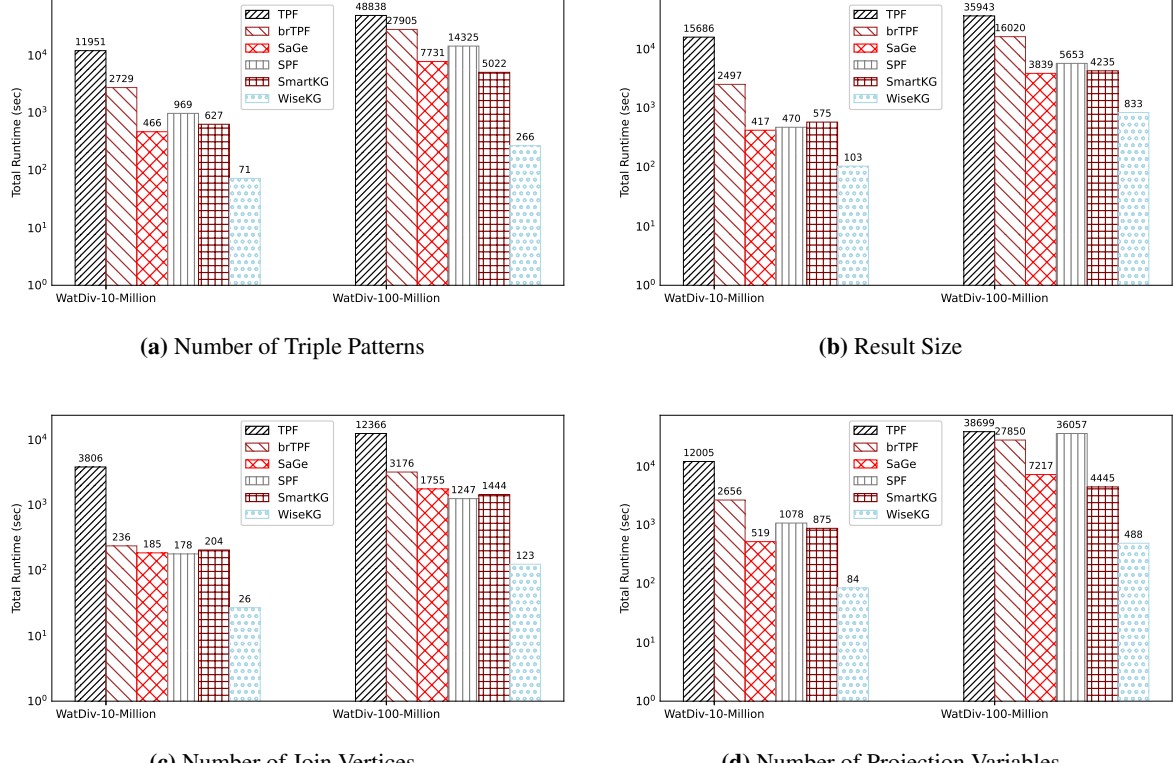

**(a)** Number of Triple Patterns

**(b)** Result Size

**(c)** Number of Join Vertices

**(d)** Number of Projection Variables

**Figure 4:** Total runtime in seconds observed for all querying interfaces on both *WatDiv-10-Million* and *WatDiv-100-Million* datasets when executing queries with varying query features: triple patterns, result size, join vertices, and projection variables.

conditions. The intermediate performers—SaGe, SPF, and Smart-KG—displayed closely comparable runtimes, with only minor variations between them. A notable exception occurred with SPF on the projection variable (PV) queries against the 100 million triple dataset, where its performance degraded significantly to match TPF's execution time.

Importantly, no query timeouts were encountered during these experiments, as we configured sufficiently generous execution time limits to capture complete runtime profiles for all queries and interfaces. Based on the total runtime measurements, the performance ranking from best to worst is as follows: Wise-KG (1st), Smart-KG (2nd), SaGe (3rd), SPF (4th), brTPF (5th), and TPF (6th). These aggregate findings align with the average runtime patterns observed in Figures 2 and 3, where per-query-group measurements confirm the overall interface hierarchy while providing additional granularity regarding query feature sensitivity. This consistent ranking across multiple analysis perspectives strengthens confidence in the robustness of our comparative evaluation.

## 7. Resource Availability and Reusability

The datasets and queries used in this study are based on the widely recognized WatDiv benchmark [3]. To tailor the evaluation to specific query characteristics, we enriched the queries using LSQ [35], allowing precise selection based on key structural features such as triple patterns, projection variables, join vertices, and result set sizes, as described in Section 3.

All six SPARQL querying interfaces were deployed and configured following the official guidelines and documentation provided by their respective maintainers. Care was taken to ensure consistent setup, fair comparisons, and uniform query execution procedures across all systems. The datasets used in our experiments include both 10 million and 100 million triple versions of the WatDiv dataset, allowing us to observe system behavior at different scales. An overview of dataset details is presented in Table 2.

To ensure transparency and reproducibility, the complete project repository is publicly available online[4]. The repository includes all components necessary to replicate or extend the experiments: the full set of selected queries, the configuration and execution scripts for each interface, the resulting performance metrics, and the datasets used.

By making our experimental setup and results openly accessible, we aim to support further research and foster reproducible evaluations of SPARQL querying interfaces. Researchers can not only reproduce our findings but also use the provided resources to benchmark new systems, explore additional query features, or validate cost estimation models.

## 8. Conclusion and Future Work

We performed a comprehensive analysis of existing *SPARQL Querying Interfaces* by executing different sets of queries having different features, i.e., *TP*, *RS*, *JV* and *PV* by using *WatDiv* as benchmark dataset with two different sizes. Our evaluation results suggest the following:

(1) Runtime performance and scalability are not correlated, as we see TPF performed the worse in case of query runtime performance but it is the most scalable and has observed very less effect when exposed to a larger size dataset. On the other hand, SPF performed very well in terms of query runtime performance but showed worse results in terms of percentage difference in runtimes when exposed to a larger sized dataset.

(2) Runtime performance and all the query features we considered for our experiments, i.e., TP, RS, JV and PV, has correlation between them. However in case of WatDiv-10M all showed *very strong* correlation except PV, which showed *strong*. On the other hand, with WatDiv-100M dataset all showed *strong* correlation but TP showed *very strong* on average.

(3) Total runtime is seconds, of all the querying interfaces was calculated by executing all sets of the queries, and with both sizes of the datasets.

As future work, we endeavour to broaden the scope of our analysis by adding more types of SPARQL query benchmarks, more query features and more evaluation metrics.

## Acknowledgements

This section will be completed in the camera-ready version.

## Declaration on Generative AI

During the preparation of this work, the author(s) used Generative AI tools including ChatGPT in order to improve grammar, spelling, and language clarity. The author(s) reviewed and edited the content as needed and take full responsibility for the content of the publication.

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
