# OpenReview forum: "How do Query Features correlate with Runtime? An evaluation of SPARQL Querying Interfaces"
_SEMANTiCS.cc/2026/Workshop/UKG — SEMANTiCS 2026 Workshop UKG Submission_

### Official Review · ~Davan_Chiem_Dao1 · 2026-07-09
**Review of "How do Query Features correlate with Runtime? An evaluation of SPARQL Querying Interfaces"**

**Rating:** 5
**Confidence:** 3

**Review:**

## Summary
The paper presents an empirical performance evaluation of six client-server SPARQL querying interfaces (TPF, brTPF, SaGe, SPF, Smart-KG, Wise-KG). The authors investigate how SPARQL query features—such as the number of triple patterns (TP), join vertices (JV), projection variables (PV), and result set sizes (RS)—correlate with execution time. Using the WatDiv benchmark at two different dataset scales (10M and 100M triples), the authors calculate Spearman’s rank correlation coefficients to quantify the impact of query complexity on runtime and show the correlation between them.

## Strengths
- The paper is well written and resources/examples are made available for reproducibility.
- The use case is relevant, as the evaluation of client-server SPARQL interfaces is a highly relevant topic for the Semantic Web community.

## Weaknesses (Major Issues)
- Limited contribution: Demonstrating that query features (like the number of triple patterns or result set size) correlate with runtime is somewhat trivial. It is expected that larger, more complex queries take longer to execute. The paper establishes that correlations exist but does not explain why specific query features increase runtime for certain interfaces, which would be highly valuable.
- Unclear and potentially unfair experimental setup: Despite claims of a fair comparison, the interfaces do not all use the same underlying server implementation (e.g., not all use the LinkedDataFragments server), which can skew the runtime results due to server differences. Furthermore, the use of the Comunica client could introduce another layer of bias. To the best of my knowledge, Comunica performs complex client-side query planning, which may inherently favor certain server architectures over others, making the comparison less about the interfaces themselves and more about Comunica's optimization strategies. Using a more basic client might reveal runtime differences that are strictly due to architectural differences.
- Linear scalability claim: The authors claim that runtime performance scales approximately linearly with dataset size. This claim should be significantly relaxed or removed, as evaluating only two dataset sizes is insufficient to prove any linear scaling trend.

## Minor Comments
- Missing citation: The reference for LSQ should be added the very first time it appears in the text (e.g., "From the LSQ-enriched set of generated queries...").
- Consistency vs performance: The conclusion regarding TPF is somewhat unconvincing. The authors highlight TPF's "consistency" across dataset sizes as a positive, but it is difficult to imagine a practical real-world scenario where runtime consistency outweighs absolute performance, especially when TPF is drastically slower than the alternatives.
- Data discrepancy: There appears to be an inconsistency in the reported results. If one divides the WatDiv-100M results by the WatDiv-10M results shown in Figure 4, the resulting ratios do not match the percentage values reported in Table 1. Could you check and clarify?
-
## Overall Assessment

Overall, this paper tackles a highly relevant topic for the Semantic Web community and provides a well-written, reproducible benchmarking effort. However, the overall contribution feels somewhat limited, as the correlation between query complexity and runtime is largely expected. The paper would be significantly stronger if it offered deeper insights into why these correlations exist for specific interfaces, rather than just proving that they do. Additionally, the authors should to address the potential biases in their experimental setup.

---

### Official Review · ~Eduard_Kamburjan1 · 2026-07-10

**Rating:** 5
**Confidence:** 3

**Review:**

This paper evaluates how different SPARQL query features influence the runtime of different SPARQL query interfaces.
This generalizes prior work that evaluates SPARQL endpoints.

The work fits the scope of the workshop and is generally well-written, but I have some concerns about the setup of the experiments.
1. The authors choose the 4 features that had the strongest influence for endpoints, but this excludes the possibility to detect that some features have a bigger influence for query interfaces.
2. It is ultimately unclear what question the authors aim to answer. This is mainly because the paper does not formulate explicit research questions, instead it reruns experiments targeting a different interface than in prior work and report that all features have a strong influence, and that for all performance metrics the more complex interfaces perform better. I think the aim was exactly to confirm that evaluating the runtime for endpoints and query interfaces shows no differences, but it would be good to have that states explicit.

Similarly to the last point, the paper does not discuss threats to validity or provide insights. The conclusion is rather shallow: (2) and (3) are just repeating data, while (1) follows directly from it. The overall trend that more complexity=more runtime is also clear. What are the insights the community gains from this evaluation?

As some minor points:
 - The tables and figures in Sec. 6 are following a different order in discussion and placement, e.g., Fig. 4 is discussed before Fig. 2.
 - Table 2 would be more intuitive if the numbers X were reported as +X%
 - Please check the bibliography, [1] is all upper case for the authors, several publications (e.g., [15-20]) lack DOIs
 - Given that only one data set was used, how generalizable are these results?

I think the paper can lead to interesting discussions about evaluation of KG interfaces, but should frame its empirical investigations in a more structured form.

---

### Official Review · ~Ademar_Crotti_Junior1 · 2026-07-22
**The paper presents experiments on client-server sparql query interfaces**

**Rating:** 6
**Confidence:** 3

**Review:**

Summary:
The paper presents experiments on client-server sparql query interfaces.

Strengths:
- It uses the well-known benchmark WatDiv
- Understanding how these engines work is beneficial for improving them, and also deploying them depending on use cases and kinds of queries expected
- All experiment data is available.

Weaknesses:
- I think you can make it clear in the introduction what it is that your evaluation is trying to answer.
- The conclusion is that complexity correlates to runtime. A more interesting aspect would be to consider why one engine performs better than another for the same data and query, as well as same engine architecture. For example: how do these engines differ in terms of implementation, the number of http requests, intermediate result sets, payload, client side join, etc. You provide some description of the evaluated engines in section 2, some of those characteristics could be used later to try identify why the different engine performances and how they can learn from each other.
- You mention this as a weakness of prior work in the introduction "no existing study has quantified the relationship between query features and runtime across the full range of SPARQL querying interfaces", yet I am not sure you did.